# Influence of pharmacists and infection control teams or antimicrobial stewardship teams on the safety and efficacy of vancomycin: A Japanese administrative claims database study

**Ryota Goto[1], Yuichi Muraki[1]*, Ryo Inose[1], Yoshiki Kusama[2], Akane Ono[3], Ryuji Koizumi[3], Masahiro Ishikane[3], Norio Ohmagari[3]**

1 Department of Clinical Pharmacoepidemiology, Kyoto Pharmaceutical University, Kyoto, Japan, 2 Division of General Pediatrics, Department of Pediatrics, Hyogo Prefectural Amagasaki General Medical Center, Amagasaki, Hyogo, Japan, 3 AMR Clinical Reference Center, Disease Control and Prevention Center, National Center for Global Health and Medicine, Tokyo, Japan

* y-muraki@mb.kyoto-phu.ac.jp

**Data Availability Statement:** In this study, data owned by Medical Data Vision (MDV) are acquired

## Abstract

### Introduction

Methicillin-resistant *Staphylococcus aureus* (MRSA) has a high mortality and requires effective treatment with anti-MRSA agents such as vancomycin (VCM). Management of the efficacy and safety of VCM has been implemented with the assignment of pharmacists in hospital wards and the establishment of teams related to infectious diseases. However, there are no reports evaluating the association between these factors and the efficacy and safety of VCM in large populations.

### Methods

This study used the Japanese administrative claims database accumulated from 2010 to 2019. The population was divided into two groups, therapeutic drug monitoring (TDM) group and non-TDM group, and adjusted by propensity score matching. We performed multivariate logistic regression analysis to determine the influence of pharmacists and infection control teams or antimicrobial stewardship teams on acute kidney injury (AKI) and 30-day mortality.

### Results

The total number of patients was 73 478 (TDM group, n = 55 269; non-TDM group, n = 18 209). After propensity score matching, 18 196 patients were matched in each group. Multivariate logistic regression analysis showed that pharmacological management for each patient contributed to the reduction of AKI (odds ratio [OR]: 0.812, 95% confidence interval [CI]: 0.723–0.912) and 30-day mortality (OR: 0.538, 95% CI: 0.503–0.575). However, the establishment of infectious disease associated team in facilities and the assignment of pharmacists in the hospital wards had no effect on AKI and 30-day mortality. In addition, TDM

under contract. Therefore, the data are not available to the public. The authors have no special privileges to access the data. If other researchers wish to use the data, they must purchase it from MDV as well as the authors. To contact MDV, please use the following website (https://www.mdv.co.jp/).

**Funding:** This work was supported by the Ministry of Health, Labor, and Welfare (grant number 20HA2003).

**Competing interests:** Yuichi Muraki received an honorarium for lecturing from Pfizer Japan Inc. The other authors have no conflicts of interest to declare.

did not affect the reduction in AKI (OR: 1.061, 95% CI: 0.948–1.187), but reduced 30-day mortality (OR: 0.873, 95% CI: 0.821–0.929).

## Conclusion

Pharmacologic management for individual patients, rather than assignment systems at facilities, is effective to reduce AKI and 30-day mortality with VCM administration.

## Introduction

It is estimated that more than 10 million people worldwide will die from infections caused by drug-resistant bacteria by 2050 [1]. Among them, methicillin-resistant *Staphylococcus aureus* (MRSA) is the most frequent antibiotic-resistant bacterium isolated in hospitals [2]. In a Japanese report, MRSA accounted for 24.6% of deaths due to *Staphylococcus aureus* bacteremia [3]. Therefore, it is necessary to promote the appropriate treatment for MRSA to reduce the mortality.

We previously studied the trends of anti-MRSA agents utilized in Japan using sales data and clarified that vancomycin (VCM) accounted for 50% of the total usage from 2006 to 2015 [4]. It is important to enhance the efficacy and safety of VCM because it is administered in higher proportions than other anti-MRSA agents. VCM has acute kidney injury (AKI) as a side effect [5], and AKI has been reported to be associated with increased mortality [6]. Accordingly, the guidelines for implementation of antimicrobial stewardship programs published by the Infectious Diseases Society of America and the Society for Healthcare Epidemiology of America recommend pharmacokinetic monitoring and adjustment programs for VCM in hospitals to avoid the AKI [7]. Therefore, therapeutic drug monitoring (TDM) is recommended to measure the drug concentration in the blood and analyze dosing plans for each patient [8].

In Japan, the assignment of pharmacists to hospital wards and the establishment of infection control teams (ICT) and antimicrobial stewardship teams (AST) are being promoted to manage the efficacy and safety of drugs including VCM. A previous single institutional study reported that pharmacist intervention reduced AKI or mortality in patients treated with VCM [9]. However, there is a lack of large-scale reporting on pharmacist interventions. In addition, the influence of ICT and AST interventions on AKI and mortality with VCM has not yet been clarified.

These issues can be clarified using information on the progress of drug treatment and reimbursement (S1 Table). In Japan, reimbursement is established for the medical system and prescribing behavior due to the public insurance system [10]. The claims database, which comprises real-world data, includes information obtained from multiple institutions over a long period of time [11]. Therefore, it is possible to evaluate the outcomes of various interventions for a large population. Nevertheless, a study using this approach has not been reported previously. In this study, we examined the factors affecting the efficacy and safety of VCM, while considering the interventions of pharmacists, ICTs and ASTs.

## Methods

### Study design

This retrospective study was conducted using the Japanese administrative claims data obtained from the Medical Data Vision database between 2010 and 2019. This database consists of

information from hospitals where the diagnosis procedure combination (DPC) system, mainly for acute inpatient care, has been introduced. As of 2019, it included the data from approximately 22% of all the hospitals in Japan that implement DPC systems. This database has been used previously for various studies in Japan [12,13].

In this study, the following four types of data files were analyzed: patient data containing information on the facility where the patient was treated, disease data comprising diagnostic information, medical practice data including information on treatments and procedures, and discharge summary data constituting information on injuries and sickness and various scores at the time of admission and discharge.

## Study population

The patients were classified into two groups depending on TDM implementation (TDM and non-TDM groups). Considering TDM is generally performed in hospitalized patients, only these patients who received VCM injections were included in this study. We identified the first time each patient was hospitalized for at least three consecutive days while VCM treatment was initiated between 2010 and 2019. Patients were excluded if any of the required data, such as weight or discharge information, were missing or if AKI was recorded in the month prior to VCM initiation.

The treatment and management fee for specific drugs is calculated in cases where drugs that are recommended for TDM, such as glycopeptides including VCM, some immunosuppressants, or antiepileptic drugs, are administered. Therefore, when several drugs requiring TDM were administered, it was impossible to identify the fee for which drug. Patients who received drugs requiring TDM other than VCM within 7 days of the index date (the date of initiation of VCM treatment) were excluded (S2 Table).

## Definitions and clinical variables

Based on the number of beds, facilities were categorized into the following groups: $\leq 199$, 200–499, and $\geq 500$. Comorbidity was assessed using the Charlson Comorbidity Index (CCI). We identified claims for reimbursement related to infections and drug treatment (S1 Table). Information on TDM implementation is not included in the claims database. Therefore, similar to the method applied in another study [14], we substituted it with the treatment and management fee for specific drugs, calculated once a month when TDM was conducted for the target drug. As the previous study described, nephrotoxic drugs were classified into the following groups: angiotensin II receptor blockers/angiotensin-converting enzyme inhibitors, diuretics, liposomal amphotericin B, nonsteroidal anti-inflammatory drugs, steroids, and piperacillin/tazobactam [15], and were extracted when administered on the same day as that of VCM. The site of infection was classified based on the indications related to the infection in the diagnostic information (S3 Table).

AKI was defined based on the International Classification of Diseases 10th version (ICD-10) codes (N17X: acute renal failure) and five Japanese disease codes (5839017: kidney injury, 8835584: reduced renal function, 8840719/9952036: drug-induced kidney injury, 8849597: acute kidney injury [AKI]). A previous study in elderly subjects had shown that ICD-10 code N17X for AKI had moderate sensitivity [16]. Therefore, defining AKI using ICD-10 codes alone may not be sufficient, and the five Japanese disease codes mentioned above were also defined in conjunction with N17X in this study. The data of only the patients with a confirmed diagnosis of AKI was extracted, and patients with only a suspected diagnosis were not included.

Finally, the study period for AKI incidence was defined as the period from the index date to 7 days after completion of administration, and the 30-day mortality was defined as death recorded within 30 days of the index date.

## Statistical analysis

Propensity scores were calculated by logistic regression analysis and 1:1 matching was performed with a caliper distance of 0.2. In this study, propensity score matching was performed using the following variables as factors related to the implementation of TDM; number of beds, CCI, infectious disease-associated teams fee, inpatient pharmaceutical service premium, respiratory infection, bacteremia/sepsis, urinary tract infections, intra-abdominal infections, skin and soft tissue infections, bone and joint infections, central nervous system infections, febrile neutropenia, infective endocarditis, and MRCNS infection. The variable balance between the two groups after propensity score matching was evaluated by standardized mean difference (SMD). A SMD < 0.10 suggests an appropriate variable balance. In addition, univariate and multivariate logistic regression analyses were performed with each as the objective variable in order to clarify the risk factors for AKI and 30-day mortality.

Statistical analysis was performed using Stata software, version 17.0 (Stata Corp., College Station, TX) and EZR (Saitama Medical Center, Jichi Medical University, Saitama, Japan) [17], with a significance level set at p < 0.05.

This study was approved by the ethics committee of the Kyoto Pharmaceutical University (approval number: E21-014). The requirement for informed consent was waived owing to the anonymized nature of the study data.

## Results

### Patient characteristics

Fig 1 illustrates the patient-selection process. The overall study population comprised 146 542 patients who received VCM at the time of admission; 73 478 patients were included in the study according to the patient-selection criteria.

Table 1 shows the characteristics of the study participants before and after propensity score matching. The data of 73 478 patients (TDM group, n = 55 269; non-TDM group, n = 18 209) were examined in this study. Before propensity score matching, the population differed in number of beds, infectious disease-associated teams fee, bacteremia/sepsis, and inpatient pharmaceutical service premium (SMD > 0.100). After propensity score matching, 18 196 patients were matched in each group. As a result, patient characteristic balanced the populations (SMD < 0.100), except for bed size (SMD = 0.200).

### Risk factors for AKI with VCM administration

Risk factors for AKI were examined by multivariate logistic regression analysis (Table 2). Clinical variables that increased the incidence of AKI may include men (OR: 1.219, 95% CI: 1.084–1.371) and bacteremia/sepsis (OR: 1.863, 95% CI: 1.664–2.086). On the other hand, the use of VCM for febrile neutropenia (FN) (OR: 0.691, 95% CI: 0.513–0.931) decreased the risk of AKI. In terms of reimbursement, only the drug management and guidance fee which is calculated by pharmacists intervening with individual patients reduced the risk of AKI (OR: 0.812, 95% CI: 0.723–0.912). In terms of nephrotoxic drugs used during VCM administration, the use of diuretics (OR: 2.014, 95% CI: 1.795–2.258), steroids (OR: 1.295, 95% CI: 1.143–1.467), and PIPC/TAZ (OR: 1.286, 95% CI: 1.122–1.474) significantly affected AKI.

**Fig 1. Flow chart showing patient-selection process.** * Specific medication corresponds to drugs considered to be clinically eligible for TDM among those for whom treatment and management fees for specific drugs were calculated. Details are presented in S2 Table. VCM, vancomycin; TDM, therapeutic drug monitoring.

### Identifying factors affecting 30-day mortality

Multivariate logistic regression analysis was conducted to identify the factors affecting 30-day mortality (Table 3). Patient information such as men (OR: 1.239, 95% CI: 1.161–1.322), a small number of hospital beds (OR: 1.194, 95% CI: 1.059–1.346; OR: 1.236, 95% CI: 1.149–1.329), CCI (OR: 1.048, 95% CI: 1.037–1.059), AKI (OR: 1.659, 95% CI: 1.443–1.908), and administration of several anti-MRSA agents (OR: 1.255, 95% CI: 1.116–1.412), were identified as clinical variables that increased 30-day mortality. Meanwhile, treatment and management fee for specific drugs indicating TDM (OR: 0.873, 95% CI: 0.821–0.929) and drug management and

**Table 1. Baseline characteristics of patients.**

| | Before propensity score matching | | | After propensity score matching | | |
|---|---|---|---|---|---|---|
| | TDM (55 269) | Non-TDM (18 209) | SMD | TDM (18 196) | Non-TDM (18 196) | SMD |
| **Number of beds [a]** | | | | | | |
| ≤ 199 | 2 761 (5.0) | 1 588 (8.7) | 0.158 | 1 242 (6.8) | 1 575 (8.7) | 0.200 |
| 200–499 | 30 839 (55.8) | 10 228 (56.2) | | 11 991 (65.9) | 10 228 (56.2) | |
| ≥ 500 | 21 669 (39.2) | 6 395 (35.1) | | 4 963 (27.3) | 6 393 (35.1) | |
| **CCI [b]** | 2 [1, 4] | 2 [1, 4] | 0.003 | 2 [1, 4] | 2 [1, 4] | 0.058 |
| **Reimbursement [1][a]** | | | | | | |
| Infectious disease-associated teams fee | 45 947 (83.1) | 14 219 (78.1) | 0.128 | 14 289 (78.5) | 14 219 (78.1) | 0.009 |
| Inpatient pharmaceutical service premium | 28 865 (52.2) | 7 288 (40.0) | 0.247 | 6 917 (38.0) | 7 288 (40.1) | 0.042 |
| **Site of infection [a]** | | | | | | |
| Respiratory infection | 22 535 (40.8) | 7 087 (38.9) | 0.038 | 7 256 (39.9) | 7 087 (38.9) | 0.019 |
| Bacteremia/sepsis | 19 784 (35.8) | 5 628 (30.9) | 0.104 | 6 111 (33.6) | 5 628 (30.9) | 0.057 |
| Urinary tract infections | 6 307 (11.4) | 1 872 (10.3) | 0.036 | 2 090 (11.5) | 1 871 (10.3) | 0.039 |
| Intra-abdominal infections | 3 594 (6.5) | 954 (5.2) | 0.054 | 963 (5.3) | 954 (5.2) | 0.002 |
| Skin and soft tissue infections | 3 438 (6.2) | 860 (4.7) | 0.066 | 1 016 (5.6) | 860 (4.7) | 0.039 |
| Bone and joint infections | 3 192 (5.8) | 816 (4.5) | 0.059 | 978 (5.4) | 816 (4.5) | 0.041 |
| Central nervous system infections | 2 921 (5.3) | 911 (5.0) | 0.013 | 877 (4.8) | 911 (5.0) | 0.009 |
| Febrile neutropenia | 2 419 (4.4) | 788 (4.3) | 0.002 | 785 (4.3) | 787 (4.3) | 0.001 |
| Infective endocarditis | 1 212 (2.2) | 344 (1.9) | 0.021 | 398 (2.2) | 343 (1.9) | 0.021 |
| MRCNS infection | 999 (1.8) | 161 (0.9) | 0.080 | 311 (1.7) | 161 (0.9) | 0.073 |

CCI, Charlson comorbidity index; MRCNS, methicillin-resistant coagulase-negative staphylococci; SMD, standardized mean difference.

[1] Infectious disease-associated teams fee indicates patients for whom infection prevention and control premium or antimicrobial stewardship support premium was calculated. Also, details including claimed requirements for each reimbursement are shown in S1 Table.

[a] Data are expressed n (%).

[b] Data are expressed median [interquartile rate].

guidance fee indicating pharmacist intervention (OR: 0.538, 95% CI: 0.504–0.575) contributed to a reduction in the 30-day mortality. However, calculation of the inpatient pharmaceutical services premium which assign a pharmacist to a hospital ward or infectious disease-associated team fee which establish an ICTs or ASTs in a medical facility was not affected. In addition, respiratory infections (OR: 1.820, 95% CI: 1.707–1.941) and bacteremia/sepsis (OR: 1.551, 95% CI: 1.454–1.655) increased the 30-day mortality, while skin and soft tissue infections (OR: 0.430, 95% CI: 0.347–0.533) and bone and joint infections (OR: 0.301, 95% CI: 0.233–0.390) were identified as factors that decreased the 30-day mortality.

## Discussion

In this study, we evaluated the factors for AKI and mortality associated with VCM using claim data, considering the institutional background, the assignment of pharmacists in the hospital wards, and the establishment of ICTs and ASTs, which have not been examined previously.

In previous reports, TDM was evaluated for appropriate management using VCM blood concentration values [18]. On the other hand, in this study using reimbursement, although it was clear that blood concentration management was measured, the control status of the VCM concentration values was unknown. Therefore, it is possible that a decrease in the incidence of AKI due to TDM was not identified in this study.

**Table 2. Clinical variables associated with AKI in the adjusted population.**

| | Univariate Analysis | | | Multivariate Analysis | | |
|---|---|---|---|---|---|---|
| | **Crude OR** | **95% CI** | **p value** | **Adjusted OR** | **95% CI** | **p value** |
| **Patient Information** | | | | | | |
| Sex [female vs. male] | 0.839 | 0.748–0.940 | 0.003 | 1.219 | 1.084–1.371 | 0.001 |
| Age (years) | 0.998 | 0.995–1.001 | 0.149 | 0.993 | 0.990–0.997 | < 0.001 |
| Number of hospital beds | | | | | | |
| [> 500 vs. < 200] | 0.837 | 0.670–1.046 | 0.118 | 0.889 | 0.704–1.121 | 0.320 |
| [> 500 *vs.* 200–499] | 0.865 | 0.770–0.973 | 0.015 | 0.900 | 0.796–1.018 | 0.093 |
| CCI | 1.001 | 0.983–1.020 | 0.889 | 0.986 | 0.967–1.006 | 0.167 |
| **Treatment details** | | | | | | |
| Treatment category [1] [VCM only vs. medication changes] | 1.556 | 1.303–1.858 | < 0.001 | 1.460 | 1.217–1.751 | < 0.001 |
| Treatment duration (day) | 0.995 | 0.986–1.003 | 0.225 | 0.990 | 0.981–0.999 | 0.032 |
| Initial dose (mg/kg/day) | 0.994 | 0.990–0.998 | 0.003 | 0.993 | 0.989–0.997 | < 0.001 |
| **Reimbursement** [2] | | | | | | |
| Treatment and management fee for specific drugs | 1.042 | 0.935–1.162 | 0.454 | 1.061 | 0.948–1.187 | 0.301 |
| Inpatient pharmaceutical services premium | 1.015 | 0.908–1.135 | 0.791 | 1.017 | 0.905–1.144 | 0.772 |
| Drug management and guidance fee | 0.774 | 0.693–0.864 | < 0.001 | 0.812 | 0.723–0.912 | < 0.001 |
| Infectious disease-associated team fee | 1.050 | 0.919–1.200 | 0.472 | 1.044 | 0.910–1.197 | 0.538 |
| **Site of infection** | | | | | | |
| Respiratory infection | 1.068 | 0.956–1.193 | 0.244 | 1.002 | 0.890–1.128 | 0.976 |
| Bacteremia/Sepsis | 2.066 | 1.852–2.304 | < 0.001 | 1.863 | 1.664–2.086 | < 0.001 |
| Infective endocarditis | 1.293 | 0.917–1.824 | 0.142 | 1.070 | 0.754–1.517 | 0.707 |
| Skin and soft tissue infections | 1.069 | 0.842–1.358 | 0.585 | 1.260 | 0.986–1.609 | 0.065 |
| Bone and joint infections | 0.718 | 0.538–0.959 | 0.025 | 0.845 | 0.628–1.136 | 0.265 |
| Intra-abdominal infections | 1.613 | 1.319–1.973 | < 0.001 | 1.486 | 1.207–1.829 | < 0.001 |
| Central nervous system infections | 0.832 | 0.634–1.091 | 0.184 | 0.942 | 0.712–1.247 | 0.677 |
| Urinary tract infections | 1.383 | 1.183–1.617 | < 0.001 | 1.318 | 1.120–1.551 | 0.001 |
| Febrile neutropenia | 0.846 | 0.635–1.128 | 0.255 | 0.691 | 0.513–0.931 | 0.015 |
| MRCNS infection | 0.909 | 0.551–1.501 | 0.709 | 0.974 | 0.587–1.616 | 0.919 |
| **Nephrotoxic drugs used during VCM administration** | | | | | | |
| ACE-I/ARB | 0.853 | 0.718–1.014 | 0.071 | 0.852 | 0.714–1.016 | 0.075 |
| Diuretics | 2.053 | 1.841–2.290 | < 0.001 | 2.014 | 1.795–2.258 | < 0.001 |
| L-AMB | 1.623 | 1.104–2.387 | 0.014 | 1.193 | 0.802–1.777 | 0.384 |
| NSAIDs | 0.712 | 0.617–0.822 | < 0.001 | 0.789 | 0.680–0.914 | 0.002 |
| Steroids | 1.521 | 1.350–1.714 | < 0.001 | 1.295 | 1.143–1.467 | < 0.001 |
| PIPC/TAZ | 1.414 | 1.238–1.615 | < 0.001 | 1.286 | 1.122–1.474 | < 0.001 |

OR, odds ratio; CI, confidence interval; CCI, Charlson comorbidity index; VCM, vancomycin; MRCNS, methicillin-resistant coagulase-negative staphylococci.

[1] Medication change indicates patients switched from VCM to other anti-MRSA drugs and switched from other anti-MRSA drugs to VCM.

[2] Infectious disease-associated teams fee indicates patients for whom infection prevention and control premium or antimicrobial stewardship support premium was calculated. Also, details including claimed requirements for each reimbursement are shown in S1 Table.

Sepsis is defined as life-threatening organ dysfunction caused by a dysregulated host response to infection [19] and is known to cause renal damage. In the guidelines, sepsis is also described to affect AKI [20], which may reflect similar results in this patient population. On the other hand, AKI may be less likely to occur in FN. Previous reports have shown that VCM clearance is enhanced in neutropenic patients [21]. This may have resulted in a significant reduction in the risk of AKI.

**Table 3.  Clinical variables associated with 30-day mortality in the adjusted population.**

| | Univariate Analysis | | | Multivariate Analysis | | |
|---|---|---|---|---|---|---|
| | Crude OR | 95% CI | p value | Adjusted OR | 95% CI | p value |
| **Patient Information** | | | | | | |
| Sex [female vs. male] | 0.826 | 0.778–0.878 | < 0.001 | 1.239 | 1.161–1.322 | < 0.001 |
| Age (years) | 1.035 | 1.032–1.037 | < 0.001 | 1.029 | 1.027–1.032 | < 0.001 |
| Number of hospital beds | | | | | | |
| [> 500 vs. < 200] | 1.743 | 1.561–1.946 | < 0.001 | 1.194 | 1.059–1.346 | 0.004 |
| [> 500 vs. 200–499] | 1.434 | 1.340–1.534 | < 0.001 | 1.236 | 1.149–1.329 | < 0.001 |
| CCI | 1.040 | 1.030–1.050 | < 0.001 | 1.048 | 1.037–1.059 | < 0.001 |
| **Treatment details** | | | | | | |
| Treatment category [1] [VCM only vs. medication changes] | 1.040 | 0.932–1.160 | 0.486 | 1.255 | 1.116–1.412 | < 0.001 |
| Treatment duration (day) | 0.924 | 0.918–0.930 | < 0.001 | 0.915 | 0.908–0.921 | <0.001 |
| Initial dose (mg/kg/day) | 0.992 | 0.989–0.994 | < 0.001 | 1.000 | 0.998–1.003 | 0.685 |
| **Reimbursement [2]** | | | | | | |
| Treatment and management fee for specific drugs | 0.779 | 0.735–0.826 | < 0.001 | 0.873 | 0.821–0.929 | < 0.001 |
| Inpatient pharmaceutical services premium | 0.899 | 0.847–0.954 | < 0.001 | 1.047 | 0.981–1.118 | 0.170 |
| Drug management and guidance fee | 0.445 | 0.419–0.473 | < 0.001 | 0.538 | 0.504–0.575 | < 0.001 |
| Infectious disease-associated team fee | 1.007 | 0.939–1.080 | 0.838 | 0.934 | 0.867–1.007 | 0.076 |
| **Site of infection** | | | | | | |
| Respiratory infection | 2.423 | 2.285–2.569 | < 0.001 | 1.820 | 1.707–1.941 | < 0.001 |
| Bacteremia/Sepsis | 1.410 | 1.329–1.497 | < 0.001 | 1.551 | 1.454–1.655 | < 0.001 |
| Infective endocarditis | 0.823 | 0.662–1.024 | 0.081 | 0.933 | 0.741–1.176 | 0.558 |
| Skin and soft tissue infections | 0.294 | 0.239–0.362 | < 0.001 | 0.430 | 0.347–0.533 | < 0.001 |
| Bone and joint infections | 0.201 | 0.156–0.258 | < 0.001 | 0.301 | 0.233–0.390 | < 0.001 |
| Intra-abdominal infections | 0.778 | 0.676–0.894 | < 0.001 | 0.866 | 0.747–1.004 | 0.056 |
| Central nervous system infections | 0.396 | 0.328–0.478 | < 0.001 | 0.627 | 0.515–0.762 | < 0.001 |
| Urinary tract infections | 1.033 | 0.943–1.132 | 0.485 | 0.743 | 0.674–0.820 | < 0.001 |
| Febrile neutropenia | 0.833 | 0.716–0.968 | 0.017 | 1.032 | 0.878–1.212 | 0.705 |
| MRCNS infection | 0.732 | 0.551–0.974 | 0.032 | 0.835 | 0.620–1.124 | 0.234 |
| **AKI** | 1.728 | 1.517–1.969 | < 0.001 | 1.659 | 1.443–1.908 | < 0.001 |

OR, odds ratio; CI, confidence interval; CCI, Charlson comorbidity index; VCM, vancomycin; MRCNS, methicillin-resistant coagulase-negative staphylococci; AKI, acute kidney injury.

[1] Medication change indicates patients switched from VCM to other anti-MRSA drugs and switched from other anti-MRSA drugs to VCM.

[2] Infectious disease-associated teams fee indicates patients for whom infection prevention and control premium or antimicrobial stewardship support premium was calculated. Also, details including claimed requirements for each reimbursement are shown in S1 Table.

In this study, a smaller number of hospital beds increased the 30-day mortality. The treatment of infectious diseases is inadequate in facilities with a small number of hospital beds [22], which may be a factor in increasing mortality. In addition, CCI and replacement treatment were identified as factors increasing the 30-day mortality, suggesting that the disease severity affects the prognosis. However, there are reports of improved prognosis with early switching from VCM [23], and this needs to be further investigated.

As for the sites of infection, respiratory infections and bacteremia/sepsis were recognized as clinical variables related to increased mortality. Among respiratory infections, pneumonia is reported to be more common in older adults [24]. In addition, it is the fifth most common cause of death in Japan [24]. Since most patients in this study were older individuals (median age, 75 years), this might have affected the mortality. Bacteremia/sepsis is a serious infection

with a mortality proportion of over 25–30% [25] and may be a factor associated with increased 30-day mortality.

The drug management and guidance fee indicating that the pharmacist intervened with each patient contributed to the reduction in the incidence of AKI and 30-day mortality. On the other hand, no association with incidence of AKI and 30-day mortality was found for inpatient pharmaceutical services premium, which is calculated in a system with a pharmacist stationed in the hospital ward, or for infectious disease-associated team fee, which is calculated for establishing ICTs and ASTs in the hospital. The drug management and guidance fee can be claimed through participation in the treatment of individual patients by performing pharmacological management such as confirmation of drug interactions and monitoring of side effects. In contrast, inpatient pharmaceutical services premium and infectious disease-associated team fee are indicative of the facility's practices and do not indicate pharmacological management for a specific individual. Therefore, in treatment with VCM, it may be more important to intervene with individual patients than to create a facility system to reduce AKI and 30-day mortality.

There are several limitations to this study. The first is the items in the database study. In some cases, data may be missing or the data entry format may not be consistent across facilities. Also, as we are unable to identify post-discharge outcomes, there is a possibility of underestimating 30-day mortality. In addition, this population may not be representative due to sampling bias because it does not cover all patients in Japan. The data obtained is prescription information and may not reflect actual dosing. The second issue concerns definitions in this study. There is a possibility of misclassification bias due to the substitution of reimbursement for TDM and selection bias due to the exclusion of patients receiving drugs that require TDM other than VCM. These definitions may underestimate TDM implementation. Similarly, AKI was substituted for ICD-10 code and Japanese disease code. The incidence of AKI may also have been underestimated. In addition, the timeline of AKI diagnosis and VCM administration may have been reversed. Finally, there were unmeasured confounders including severity of infection and surgical history.

Despite these limitations, it is important to use large claim data to evaluate the impact of reimbursement on AKI and 30-day mortality with VCM administration. However, we were unable to evaluate the utility of reimbursements that do not target individuals, such as inpatient pharmaceutical services premiums and infectious disease-associated team fees. Therefore, a new reimbursement system is required for the treatment participation for individual patients. In addition, while the utilization of real-world data is being promoted, problems had been highlighted, such as the existence of receipt disease names and the lack of detailed diagnosis dates or laboratory data. In the future, medical data such as claims data should be collected with consideration for its use in research.

## Conclusion

Drug management and guidance fees for each patient reduced AKI and 30-day mortality with VCM. On the other hand, inpatient pharmaceutical services premiums and infectious disease-associated team fees by system at the facility did not affect AKI and 30-day mortality. In order to decrease AKI and 30-day mortality, pharmacist interventions for individual patients need to be further promoted.

## Supporting information

**S1 Table. Reimbursement and claimed requirements.** TDM, therapeutic drug monitoring; AEDs, antiepileptic drugs; ICT, infection control team; AST, antimicrobial stewardship team. (DOCX)

**S2 Table. Number of patients who claimed treatment and management fee for specific drugs excluding vancomycin.** * "Number" indicates the number of patients who used the drug concerned concomitantly within 7 days from the index date, defined as the date of vancomycin initiation.
(DOCX)

**S3 Table. Codes and diagnostic information for classification of infection sites.** These codes were defined based on the Various Information of Medical Fee operated by the Ministry of Health, Labour and Welfare (https://shinryohoshu.mhlw.go.jp/). MRCNS, methicillin-resistant coagulase-negative staphylococci.
(DOCX)

## Acknowledgments

We would like to thank Editage (https://www.editage.com) for English language editing.

## Author Contributions

**Conceptualization:** Ryota Goto, Yuichi Muraki, Ryo Inose, Yoshiki Kusama, Akane Ono, Ryuji Koizumi, Masahiro Ishikane, Norio Ohmagari.

**Data curation:** Ryota Goto.

**Formal analysis:** Ryota Goto.

**Funding acquisition:** Yuichi Muraki, Norio Ohmagari.

**Investigation:** Ryota Goto.

**Methodology:** Ryota Goto, Yuichi Muraki, Ryo Inose.

**Project administration:** Yuichi Muraki.

**Resources:** Yuichi Muraki.

**Software:** Ryota Goto.

**Supervision:** Yuichi Muraki.

**Validation:** Ryota Goto, Ryo Inose.

**Visualization:** Ryota Goto.

**Writing – original draft:** Ryota Goto.

**Writing – review & editing:** Yuichi Muraki, Ryo Inose, Yoshiki Kusama, Akane Ono, Ryuji Koizumi, Masahiro Ishikane, Norio Ohmagari.

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
