## [Decision Letter · Decision Letter 0]

2 Jun 2022

PONE-D-22-09316Effect of therapeutic drug monitoring on the incidence of acute kidney injury and 30-day mortality after vancomycin administration: a Japanese administrative claims database studyPLOS ONE

Dear Dr. Muraki, 

Thank you for submitting your manuscript to PLOS ONE. After careful consideration, we feel that it has merit but does not fully meet PLOS ONE’s publication criteria as it currently stands. Therefore, we invite you to submit a revised version of the manuscript that addresses the points raised during the review process.

We look forward to receiving your revised manuscript.

Kind regards,

Tze Shien Lo, MD

Academic Editor

PLOS ONE

Journal Requirements:

Additional Editor Comments (if provided):

Dear Dr. Muraki,

Thank you very much for submitting your manuscript to Plos One. The reviewers recommend reconsideration of your manuscript following major revision. I invite you to resubmit your manuscript after addressing the three reviewers' comments. When revising your manuscript, please consider carefully all issues mentioned in the comments. Please respond to the reviewers' comments in a point by point manner.

Reviewers' comments:

Reviewer's Responses to Questions

**Comments to the Author**

1. Is the manuscript technically sound, and do the data support the conclusions?

Reviewer #1: Partly

Reviewer #2: No

Reviewer #3: No

2. Has the statistical analysis been performed appropriately and rigorously? 

Reviewer #1: No

Reviewer #2: No

Reviewer #3: N/A

3. Have the authors made all data underlying the findings in their manuscript fully available?

Reviewer #1: No

Reviewer #2: Yes

Reviewer #3: Yes

4. Is the manuscript presented in an intelligible fashion and written in standard English?

Reviewer #1: No

Reviewer #2: Yes

Reviewer #3: Yes

5. Review Comments to the Author

Reviewer #1: Major comments

1. The retrospective study investigated the implementation of TDM over time and the effect of TDM on AKI and mortality. However, there has been relative solid evidence that therapeutic drug monitoring of vancomycin is associated with decreased risk of acute kidney injury. The rationale to perform this study requires further clarification.

The Introduction section stressed that the impact of pharmacist and ICT/AST intervention on VCM’s prognosis and side effects hasn’t been clarified. However, the study didn’t clarify this issue either, as I didn’t find that the study considered the intervention of pharmacists and infectious disease-associated teams adequately (only table 1 and post-hoc analysis on 30-day mortality mentioned intervention of pharmacists and infectious diseases-associated teams).

2. Line 102-105: The rationale of substitution TDM with treatment and management fee is not justified adequately, which may be vulnerable from misclassification bias. I was wondering could you provide more details on this issue, including how pharmacist and infectious disease – associated team intervention was identified.

3. Figure 1: Please check the accuracy of number in Fig 1 again, especially in the first 2 steps.

4. The language of this manuscript requires major improvement, especially in Abstract section.

Minor comments:

Abstract:

1. The Introduction was overlong, and I don’t think that the contents of this study took account of interventions of pharmacists and infectious disease-associated teams adequately.

2. Methods: “The Japanese administrative claims database was used to evaluate the incidence of acute kidney injury (AKI) and 30-day mortality after VCM administration from 2010 to 2019 based on TDM implementation” I was wondering why the trend of acute kidney injury incidence and 30-day mortality from 2010 to 2019 was set as the primary analysis, which didn’t match to the title of this manuscript.

3. Results: More details (number, rate, ratio) of quantitative analysis should be added, rather than narrative description alone.

Introduction

4. Line 84-86: I get it that the study at first investigated the implementation of vancomycin TDM over time, and then evaluated the effect of TDM on mortality and AKI. However, the consideration of the interventions of pharmacists and infectious disease-associated teams was inadequate. Could you please rephrase this part?

Methods:

Statistical Analysis

5. Line 152-157: The primary analysis didn’t match with the objective/title of this study. I was wondering why the linear regression was performed on the trend of AKI incidence and 30-day mortality in both TDM and non-TDM group.

Rather, the multivariable logistic regression was used as post-hoc analysis. I am not sure whether the authors adjust confounders properly.

I suggest to add the post-hoc multivariable analysis on AKI.

Results:

6. Table 1: I was wondering could you please add the baseline characteristics of patients in the TDM group and non-TDM group, which may help interpret the results.

7. Line 207: I didn’t get it why the initial VCM dose was analyzed, as initial VCM dose may be less relevant to TDM implementation. The change and maintain dose following TDM may be more clinically relevant.

8. Line 238: “increased” should be removed from the table title.

9. The results section should be re-organized and make it more logically.

Discussion:

9. Line 261-262: the unit should be μg/mL

10. Line 284-285: I am not sure whether the study assessed “TDM and interventions by pharmacists and infectious disease-associated teams” as a whole.

11. Line 298-300:

Did the study adjust for measured confounders properly? Also, pleased provide examples of important unmeasured confounders

Please add potential selection bias (Patients who received drugs requiring TDM other than VCM within 7 days of the index date were excluded)

I am not sure whether the method of using reimbursement to identify TDM-related intervention would be vulnerable from misclassification bias.

One of the strengths of claim data may be long follow-up, which allow us to assess TDM on 3-day mortality.

Reviewer #2: The aim of this study is important, but its scientific validity is not high. I would like to re-read the resubmitted content, as I think considerable revisions are necessary.

Major comments

Are the diagnosis names of AKI (e.g., ICD-10 codes) validated? Do these AKI diagnostic names reflect true AKI? The percentage of AKI in this study is clearly lower than in previous reports.

How did the authors handle suspected diagnosis? Clinicians often make a suspected diagnosis to levy medical fees for performing blood tests and prescriptions in Japan.

Why did the authors exclude CKD patients (N18X)?

Usually, there is a difference in patient background between patients with TDM and those without. The authors should confirm this difference and then evaluate outcomes after adjusting for patient background (e.g., propensity score matching).

Minor comments

Figure 1 has two calculation errors.

The limitation is inadequate. For example, a discussion of sampling bias resulting from the employing DPC data is needed. The authors are only evaluating prescriptions, not the actual dosing situation.

The authors should recognize the difference between rate, proportion, and ratio and use them appropriately.

Reviewer #3: I read with interest of your manuscript entitled “Effect of therapeutic drug monitoring on the incidence of acute kidney injury and 30-day mortality after vancomycin administration: a Japanese administrative claims database study”. I understand TDM is important for adequate dose change for vancomycin, However, authors may not conclude relation between TDM implementation and AKI using administrative data.

Major

1. Authors defined for AKI as primally endpoint such as “AKI was diagnosed based on the ICD-144 10 code (N17X) and five Japanese disease codes (5839017: kidney injury, 8835584:145 reduced renal function, 8840719/ 9952036: drug-induced kidney injury, 8849597: AKI).”

Definition for AKI using N17X is reported for inadequate by the results of validation analysis in renal transplantation patients 1, 2) In addition, Japanese disease codes for authors definition may not be adequate to detect AKI.

Ref 1) JSPE committee, reports available from “https://www.jspe.jp/committee/pdf/

validationtrr120180523.pdf” (see p.27, line 16)

Ref 2) Amber O. Molnar etal. Validation of administrative database codes for acute kidney injury in kidney transplant recipients. Can J Kidney Health Dis. 2016; 3: 18.

2. Detailed date for AKI is able to define by using DPC data? Is the time axis reversed between AKI diagnosis and VCM administration?

3. How to analyze “30-day mortality after VCM administration.”? Authors can assess death after the discharge? Administrative data from Medical Data Vision can provide DPC (in-hospital) and the after discharge?

Minor

1. Authors calculated “rate of TDM implementation” Is it correct? Is it proportion? (line 149)

6. PLOS authors have the option to publish the peer review history of their article (what does this mean?). If published, this will include your full peer review and any attached files.

Reviewer #1: No

Reviewer #2: No

Reviewer #3: No

---

## [Author Response · Author response to Decision Letter 0]

1 Aug 2022

To editor and reviewers

Thank you very much for taking the time to review our paper. 

We appreciate your helpful suggestions on many issues, including adjustments to the patient background, 

various definitions and biases, and the organization of the manuscript. 

In response to your comments, we have substantially revised the title and content of the paper. 

Our corrections and comments are as follows documents. 

We would appreciate if you could confirm them.

---

## [Editor Report · Decision Letter 1]

26 Aug 2022

Influence of pharmacists and infection control teams or antimicrobial stewardship teams on the safety and efficacy of vancomycin: a Japanese administrative claims database study

PONE-D-22-09316R1

Dear Dr. Yuichi Muraki,

We’re pleased to inform you that your manuscript has been judged scientifically suitable for publication and will be formally accepted for publication once it meets all outstanding technical requirements.

Kind regards,

Tze Shien Lo, MD

Academic Editor

PLOS ONE

Additional Editor Comments (optional):

Dr. Muraki and his co-workers have responded to the reviewers' comments well and have made appropriately changes recommended by the reviewers. Therefore, the revised manuscript has met PLos ONE's Publication Criteria.
---

## [Editor Report · Acceptance letter]

30 Aug 2022

PONE-D-22-09316R1 

Influence of pharmacists and infection control teams or antimicrobial stewardship teams on the safety and efficacy of vancomycin: a Japanese administrative claims database study 

Dear Dr. Muraki:

I'm pleased to inform you that your manuscript has been deemed suitable for publication in PLOS ONE. Congratulations! Your manuscript is now with our production department. 

Kind regards, 

on behalf of

Dr. Tze Shien Lo 

Academic Editor

PLOS ONE